# Feline Uroepithelial Cell Culture as a Novel Model of Idiopathic Cystitis: Investigations on the Effects of Norepinephrine on Inflammatory Response, Oxidative Stress, and Barrier Function

**DOI:** 10.3390/vetsci10020132

**Published:** 2023-02-08

**Authors:** Patrícia Hatala, Andrea Lajos, Máté Mackei, Csilla Sebők, Patrik Tráj, Júlia Vörösházi, Zsuzsanna Neogrády, Gábor Mátis

**Affiliations:** Division of Biochemistry, Department of Physiology and Biochemistry, University of Veterinary Medicine, István Utca 2., H-1078 Budapest, Hungary

**Keywords:** feline idiopathic cystitis, urinary bladder, norepinephrine, pro-inflammatory cytokines, oxidative stress, bladder barrier, SDF-1, claudin-4, TER, GAG

## Abstract

**Simple Summary:**

Feline idiopathic cystitis is a common disease in domestic cats. The pathogenesis of the illness is not completely understood, but the role of various stress factors and the concomitant release of stress hormones, such as norepinephrine, is strongly suggested. Therefore, the examination of this hormone is essential to gain a deeper knowledge of the development of the disease. In the present study, a novel cell culture of uroepithelial cells from a feline urinary bladder was established to serve as a proper model for studying the effects of a norepinephrine triggered stress reaction. Acute, 1 h norepinephrine exposure affected uroepithelial cells by increasing metabolic activity, inducing a proinflammatory response, triggering oxidative stress, and decreasing the barrier integrity of the cultured cells. The results of this study underline that stress-associated norepinephrine release has a direct molecular effect on the uroepithelial cells; the reaction of these cells may play an important mediatory role in the pathogenesis of the disease. The established cell culture model can be a good tool for further in vitro investigations related to urinary disorders of cats.

**Abstract:**

Feline idiopathic cystitis (FIC) is one of the most common urinary tract disorders in domestic cats. As stress is suggested to play a key role in the pathogenesis of FIC, the effects of norepinephrine (NE) as a stress mediator were investigated on a novel feline primary uroepithelial cell culture, serving as an in vitro model of the disease. The uroepithelial cells gained from the mucosa of the bladder of a euthanized cat were cultured for 6 days and were acutely exposed to NE (10, 100, and 1000 µM) for 1 h. NE increased the metabolic activity of the cultured cells and elevated the extracellular concentrations of the pro-inflammatory mediators interleukin-6 (IL-6) and stromal cell derived factor 1 (SDF-1), confirming that NE can trigger an inflammatory response in the uroepithelium. Cellular protein carbonyl levels were increased by NE exposure, while malondialdehyde and glucose regulated protein 78 concentrations remained unchanged, indicating that NE may provoke the oxidative damage of proteins without inducing lipid peroxidation or endoplasmic reticulum stress. Further, it can be strongly suggested that an acute NE challenge might diminish the barrier function of uroepithelial cells, as reflected by the decreased glycosaminoglycan concentration, claudin-4 protein expression, and reduced TER values of the NE-treated cell cultures. Based on these results, short-term NE exposure mimicking acute stress can provoke an inflammatory response and decrease the barrier integrity of cultured feline uroepithelial cells. Hence, it is highly expected that stress-associated NE release may play an important mediatory role in the pathogenesis of FIC.

## 1. Introduction

Lower urinary tract disease is one of the most common disorders in domestic cats (Felis silvestris catus), with symptoms including variable combinations of hematuria, stranguria, dysuria, periuria, pain, and hypersensitivity during urination. The differential diagnosis of the clinical signs may include urolithiasis, urinary tract infection, neoplasia, or parasites, but there are some cases when no specific underlying cause can be diagnosed by clinical evaluation, so the disease is often referred to as feline idiopathic cystitis (FIC). The possible cause(s) and the pathogenesis of the disorder are widely studied; FIC should be considered as a complex, multifactorial disease. It is strongly suggested that stress may have an important role in the history of FIC, thus the examination of this factor is essential to gain a deeper knowledge concerning the development of the disease [1]. FIC is very similar to the human interstitial cystitis (IC) in both its symptoms and development, therefore the examination of its pathomechanism is important not only in veterinary, but in human medicine as well, and FIC could also serve as an animal model of IC [2,3].

The special goal of the present study was to create a novel, well-characterized primary feline uroepithelial cell culture as an in vitro model, in order to study the significance of stress and other possible contributing factors in the pathogenesis of FIC. The uroepithelium consists of multiple cell layers, including a basal, a one-two layers thick intermediate, and a differentiated superficial cell layer. The latter consists of the so-called umbrella cells [4] with an asymmetrical apical plasma membrane [5], containing uroplakins as special uroepithelial differentiation products [6,7]. According to our recent knowledge, four types of uroplakins exist: Ia., Ib., II., and III., from which uroplakin III. Can be detected only in urothelial cells of the urinary bladder, ureter, and renal pelvis [8]. Therefore, uroplakin III. is suitable to characterize differentiated uroepithelial cells in cell cultures [9]. The umbrella cells are connected by tight junctions, which provide an effective barrier against ion-, ammonia-, bacteria-, and urea-flux, and make the uroepithelium suitable to form a protective barrier between the urine and blood flow. These tight junctions consist of cytoplasmic proteins, cytoskeletal elements and transmembrane proteins. The most characteristic proteins of tight junctions are zona occludens, occludin, and claudins, being suitable to indicate the barrier function of the epithelium [10]. In addition, the apical membrane of umbrella cells is covered by a layer of glycosaminoglycans (GAGs), which play a role not only in improving the barrier function but can help to prevent bacterial adhesion as well [11]. The permeability of the bladder epithelium of cats diagnosed with FIC is increased due to the diminished expression of tight junction proteins. This disruption of the bladder barrier may contribute to increased afferent nerve activity, causing bladder symptoms such as pain and hypersensitivity [2,12].

Another goal of the experiments was to examine the role of stress in the pathomechanism of the disease. Among others, the role of catecholamines should be investigated, because an increased norepinephrine (NE) level has been measured in the blood plasma of cats suffering from FIC [13,14] and in the urine of patients with human interstitial cystitis [15]. Based on the function of the hypothalamic-pituitary-adrenal axis, under physiological conditions, chronic stress induces glucocorticoid release, which inhibits the further production of catecholamines (such as that of NE) [16]. In cats with FIC, this inhibitory effect was lacking, the NE concentration of the plasma remained increased, and the size of the adrenal gland was decreased compared to healthy animals [13,14,17].

In addition to the role of the sympathetic nervous system, it is also important to study the molecular mechanisms of the inflammatory response in FIC. Based on earlier studies, the levels of the pro-inflammatory cytokine interleukin- (IL-6) and the pro-inflammatory chemokine stromal-cell derived factor-1 (SDF-1) were increased in the urine and uroepithelial cells in the case of experimentally induced cystitis in rats [18,19,20]. Further, concentrations of IL-12, IL-18, and SDF-1 in blood plasma were elevated in cats with FIC [21]. Besides the connection between inflammatory processes and the release of stress hormones, the redox state of the uroepithelial cells is also important to investigate. Inflammatory stimuli are able to produce reactive oxygen species, and the excessive production of the pro-inflammatory cytokines and chemokines may cause oxidative stress, and thereby damage cellular DNA, membrane lipids, and proteins [22]. The oxidative stress can further increase the amount of inflammatory mediators, and also the permeability of the cells due to membrane damage [23,24,25].

As the main goal of the present study was to investigate the link between stress hormone release, inflammation, oxidative stress, and epithelial barrier function, the effects of NE (as a stress hormone) were examined in the newly established uroepithelial cell culture model by assessing the cellular metabolic activity and measuring the concentrations of the inflammatory mediators IL-6 and SDF-1, the oxidative stress markers malondialdehyde (MDA), 8-hydroxy 2 deoxyguanosine (8-OHdG), protein carbonyl (PC), and glucose-regulated protein 78 (GRP78) in cell culture media and cell lysates, following an acute NE exposure. To investigate the effect of stress on barrier function, the abundance of the tight junction protein claudin-4, the glycosaminoglycan (GAG) concentration of the cell culture media, and the transepithelial electrical resistance (TER) of cultured cells were measured after NE treatment.

## 2. Materials and Methods

### 2.1. Reagents

Unless otherwise specified, all chemicals were purchased from Merck KGaA (Darmstadt, Germany). 

### 2.2. Isolation and Culturing of Epithelial Cells from Cat Bladder

A urinary bladder was obtained from a euthanized European short hair cat. The cat did not suffer from any urinary tract disease, and the cadaver was offered by the owner for scientific purposes. The study was approved by the Local Animal Welfare Committee of the University of Veterinary Medicine, Budapest on 30 March 2021. After median laparotomy, the bladder was excised and rinsed with Krebs solution (VWR, Radnor, PA, USA) supplemented with 11.1 mM glucose, pH 7.4. After carefully removing excess adipose tissue and the smooth muscle layer, the stretched bladder was placed in sterile minimal essential medium (MEM) containing 2.5 mg/mL dispase, 1% penicillin/streptomycin/fungizone, and 20 mM N-2-hydroxyethylpiperazine-N-2-ethane sulfonic acid (HEPES) at pH 7.4 and incubated overnight at 4 °C. Thereafter, the bladder was placed in a sterile Petri dish containing 20 mL trypsin-EDTA solution at 37 °C (0.25% trypsin and 1 mM EDTA). The epithelial layer was carefully scraped from the connective tissue, and the cells were dissociated by trypsinization at 37 °C for 30 min under continuous stirring. The freshly gained cell suspension was filtered through three layers of sterile gauze into a centrifuge tube and brought up to 50 mL with MEM containing 5% fetal bovine serum (FBS), 1% penicillin/streptomycin/fungizone, and 20 mM HEPES at pH 7.4, followed by centrifugation at 120× *g* for 5 min. The supernatant was carefully discarded, and the cells were resuspended in 20 mL of the same FBS-containing MEM solution and centrifuged two additional times with the same parameters. For the third centrifugation, cells were resuspended and washed in 20 mL of defined keratinocyte medium (Thermo Fisher, Waltham, MA, USA) supplemented with 0.5% gentamicin and 1% fungizone. After the final centrifugation, the viability of the cells was examined by a trypan blue exclusion test and cell counting was performed using a Bürker’s chamber. If the viability of the cells exceeded 80%, they were considered suitable for culturing, and the suspension was diluted with defined keratinocyte medium to yield the desired concentration (700,000–800,000 cells/mL).

For monitoring pro-inflammatory mediators (IL-6 and SDF-1), oxidative stress parameters (MDA, 8-OHdG, PC, GRP78), cell injury (lactate dehydrogenase, LDH), and to perform Giemsa staining, cells were plated on 24-well plates (Greiner Bio-One, Frickenhausen, Germany), and the seeding volume was set as 0.6 mL cell suspension/well. Further, 8-well Lumox x-well plates (Sarstedt, Nümbrecht, Germany; seeding volume: 0.3 mL cell suspension/well) were used for the immunofluorescent staining, and 96-well plates (Greiner Bio-One, Frickenhausen, Germany; seeding volume: 0.2 mL cell suspension/well) were applied for assessing the cellular metabolic activity and claudin-4 abundance. For measuring the TER, cells were seeded on 24-well, high-density polyester membrane inserts (Greiner Bio-One, Frickenhausen, Germany; pore size: 0.4 µm, seeding volume: 0.2 mL cell suspension in the upper chamber and 0.4 mL defined keratinocyte medium in the bottom well). All cell culture plates were previously coated with collagen type IV according to the manufacturer’s instructions. The cells were cultured for 6 days in the presence of 5% CO_2_ at 37 °C. The culture medium was changed 48 h after seeding and then every 24 h. 

### 2.3. Characterization of the Cell Culture

Cell morphology and the confluency of the cell cultures were examined by Giemsa staining, while immunofluorescent labeling was used to verify the uroepithelial origin of the cells on day 1 and 6 of culturing. For the Giemsa staining, the cells were washed with phosphate buffered saline (PBS) and fixed in 10% PBS-formalin solution at room temperature for 10 min. Then the cells were covered with Giemsa dye at room temperature for 30 min and washed with distilled water. For the latter examinations, cultures were fixed in a mixture of 30% acetone and 70% methanol at −20 °C for 10 min. The cells were washed three times with PBS and then they were incubated in a blocking solution containing 3% bovine serum albumin (BSA) in PBS at room temperature for 30 min. To confirm the epithelial origin of the cells, eFluor labeled pan-cytokeratin antibody (Cat. No. 41-9003-82, Thermo Fisher, Waltham, MA, USA) was used. The cultures were incubated in PBS solution containing 1% BSA and pan-cytokeratin antibodies in a dilution of 1:200 for 1 h at room temperature. Then, to justify the uroepithelial origin of the cells, they were stained with fluorescein isothiocyanate (FITC) labeled feline-specific uroplakin III. antibody (Cat. No. Ac-12-00200-12, Abcore, Ramona, CA, USA). The antibody was used at a 1:500 dilution dissolved in PBS containing 1% BSA, the cultures were incubated for 1 h at room temperature. Thereafter, the cells were washed with PBS, and diamidino phenylindole (DAPI) containing mounting medium was used for staining the cell nuclei. The samples were examined by an Olympus CXK-41 type microscope (OLYMPUS, Tokyo, Japan), equipped with a Canon Eos 1100D camera (Canon, Tokyo, Japan). The ImageJ software (Center for Information Technology National Institutes of Health, Bethesda, MD, USA) was used to analyze the images.

### 2.4. Norepinephrine Treatment of the Cultures

On day 6 of culturing, cells on 24-well and 96-well plates and on 24-well membrane inserts were exposed to NE dissolved in defined keratinocyte medium at concentrations of 10, 100, and 1000 µM at 37 °C for 1 h (*n* = 3/group on 24-well plates and *n* = 6/group on 96-well plates), followed by a 24 h regeneration time, and culturing without NE supplementation. Thereafter, cell culture supernatants were collected from 24-well plates, and these cells were lysed by using mammalian protein extraction reagent (M-PER) lysis buffer (Thermo Fisher, Waltham, MA, USA). For preparing the cell lysate, 300 µL M-PER reagent was added to each well of the cultures, and the cells were scraped after 5 min shaking with a sterile cell scraper and collected into Eppendorf tubes. The culture media and the lysates were stored at −80 °C until further examinations.

### 2.5. Laboratory Analyses

#### 2.5.1. Assessment of Cellular Metabolic Activity and Cell Injury

The evaluation of the metabolic activity of the cultures on 96-well plates was performed using the CCK-8 test, adding 100 μL fresh culture medium and 10 μL CCK-8 reagent to each well. The reagent contains WST-8 (water soluble tetrazolium salt), which can be reduced by cellular dehydrogenase enzymes to orange-colored formazan, and the resulting color of the media was read at 450 nm by a Multiscan GO 3.2. reader (Thermo Fisher, Waltham, MA, USA).

In order to monitor cytotoxicity, the rate of plasma membrane damage was investigated by the measurement of extracellular LDH activity with a specific enzyme kinetic photometric assay (Diagnosticum, Budapest, Hungary). First, 10 μL of culture medium was measured to 200 μL working reagent (containing 56 mM phosphate buffer, pH 7.5; 1.6 mM pyruvate, and 240 µM NADH+ H^+^). The absorbance of samples was measured at 340 nm by a Multiscan GO 3.2. reader.

#### 2.5.2. Measurement of IL-6 and SDF-1 Concentrations

IL-6 and SDF-1 concentrations were measured from both cell culture media and cell lysates. Concentrations were assayed using feline-specific IL-6 and SDF-1 ELISA kits (Cat. No. MBS085030 and MBS049100, MyBioSource, San Diego, CA, USA) according to the instructions of the manufacturer. The absorbance was read at 450 nm using Multiscan GO 3.2 reader.

#### 2.5.3. Assessment of the Redox State of the Cells

As a marker of lipid peroxidation, MDA was measured from the cell culture media with a specific colorimetric test. According to the protocol, 300 μL freshly prepared thiobarbituric acid (TBA) stock solution was mixed with 100 μL cell culture media. Solutions were incubated at 95 °C for 1 h, followed by 10 min cooling on ice. The absorbance of the samples was read at 532 nm with a Multiscan GO 3.2 reader. 

Protein damage caused by oxidative stress was examined with a Protein Carbonyl ELISA Kit (Cat. No. MBS2600294, MyBioSource, San Diego, CA, USA) by measuring the protein carbonyl content of the cell lysate. To monitor the oxidative DNA damage, 8-OHdG concentration was assayed from the cell lysate with a specific 8-OHdG ELISA kit (Cat. No. MBS808265, MyBioSource, San Diego, CA, USA). The measurements were carried out according to instructions of the manufacturer’s protocol. The absorbances of the samples were read at 450 nm by a Multiscan GO 3.2 reader.

As a marker of endoplasmic reticulum stress, a chaperone protein—GRP78—was measured from cell lysate with a feline-specific ELISA kit (Cat. No. MBS072358, MyBioSource, San Diego, CA, USA), based on the instructions of the manufacturer’s protocol. The absorbance was read at 450 nm via a Multiscan GO 3.2 reader.

#### 2.5.4. Investigation of Epithelial Barrier Function

The GAG concentration of the cell culture media was measured by a Blyscan sulfated glycosaminoglycan assay kit (Biocolor, Carrickfergus, UK). First, 50 μL of cell culture media was added to 1 mL of Blyscan dye reagent (containing 1,9-dimethylmethylene blue) and incubated for 30 min during continuous mixing, followed by centrifugation (1300× *g* for 10 min). The supernatant was carefully discarded, and the bounded dye was released from the precipitate by a dissociation reagent. Thereafter 200 μL of the samples containing dissolved dye were transferred to a 96-well microplate and the absorbances were read by a Multiscan GO 3.2 reader at 656 nm.

In order to investigate the epithelial integrity, claudin-4 content of the cells was investigated by a feline-specific colorimetric cell-based ELISA kit (Cat. No. MBS070256, MyBioSource, San Diego, CA, USA). The relative amounts of claudin-4 were measured directly in cultured cells on a 96-well plate according to the instructions of the manufacturer’s protocol. The absorbance was read at 450 nm by a Multiscan GO 3.2 reader

For the examination of the permeability of the uroepithelial cell layer, the TER measurement of cultures on 24-well membrane inserts was carried out by a EVOM2 epithelial Volt/Ohm meter (World Precision Instruments, Sarasota, FL, USA). The TER was measured directly after NE treatment, and also after 24 h of regeneration time.

### 2.6. Statistical Analysis

For statistical analysis, R 3.5.3. software (GNU General Public License, Free Software Foundation, Boston, MA, USA) was used. Differences between means were determined by one-way ANOVA, and post-hoc tests were used for pairwise comparisons. *p* < 0.05 was considered to indicate a statistically significant difference. All results are expressed as mean ± standard error of the mean.

## 3. Results

### 3.1. Characterization of the Cell Cultures

The cell morphology and the confluence of cells were examined by Giemsa staining in the case of 1- and 6-day old cultures. On day 1, cells were adhered to the plates and started to multiply (Figure 1), while, by day 6, the cell cultures grew to confluency (Figure 2). 

One day after plating, all the cells showed pan-ytokeratin positivity with eFluor labeled anti-pan-cytokeratin antibody, but no uroplakin III positivity was detected with FITC labeled antibody (Figure 3). In the case of 6-day old cultures, all the cells showed pan-cytokeratin positivity, and the majority of the cells showed uroplakin III positivity as well (Figure 4).

### 3.2. Assessment of Cellular Metabolic Activity and Cell Injury

The metabolic activity of the cells was significantly increased after 1 h 1000 µM NE treatment compared to the control group (*p* < 0.001), while no significant changes were observed in groups with lower NE concentrations (*p* = 0.371 and *p* = 0.056 for 10 and 100 µM NE, respectively) (Figure 5).

There were no significant changes in extracellular LDH activity after NE treatments compared to the non-treated control cells (*p* = 0.603, *p* = 0.807, *p* = 0.860 for 10, 100, and 100 µM NE, respectively) (Figure 6). 

### 3.3. Measurement of IL-6 and SDF-1 Concentrations

In case of 1 h 1000 µM NE treatment, a significant increase in IL-6 level was detected in the cell culture medium (*p* = 0.040), but no significant changes were observed in the extracellular IL-6 concentration in the case of 10 and 100 µM/NE treated cells compared to the non-treated control wells (*p* = 0.670, *p* = 0.665, respectively) (Figure 7). 

The concentration of SDF-1 was significantly higher in the culture media of 10, 100, and 1000 µM NE exposed cells in comparison with the controls (*p* = 0.012, *p* = 0.009, *p* < 0.001, respectively) (Figure 8).

IL-6 and SDF-1 levels of the cell lysates showed no significant changes after NE treatments (*p* = 0.432, *p* = 0.919, *p* = 0.273 for IL-6 and 10, 100, 1000 µM, respectively; *p* = 0.941, *p* = 0.624, *p* = 0.243 for SDF-1 and 10, 100, 1000 µM, respectively) (Figure 7 and Figure 8).

### 3.4. Assessment of the Redox State of the Cells

There were no significant changes in the MDA levels of cell culture media (*p* = 0.939, *p* = 0.737, *p* = 0.535, respectively) and the GRP78 abundance of cell lysate (*p* = 0.106, *p* = 0.229, *p* = 0.384, respectively) in the case of any NE treatments compared with the non-treated control cells. A significant increase in the PC level was measured in cell lysate in the case of 1000 µM NE exposure (*p* = 0.05), but no significant alterations were detected in cells challenged with 10 and 100 µM NE in comparison with the control group (*p* = 0.749, *p* = 0.689, respectively). In case of 100 µM NE treatment, a significantly lower 8-OHdG concentration was observed in the cell lysate than in the non-treated control cells (*p* = 0.019), but no significant changes were detected in the case of the other NE concentrations (*p* = 0.977, *p* = 0.828, respectively) (Figure 9).

### 3.5. Investigation of Epithelial Barrier Function

Significantly lower GAG content was observed in the medium in the case of 100 µM NE treatment (*p* = 0.008), but there were no changes when cells were exposed to 10 and 1000 µM NE in comparison with the non-treated control wells (*p* = 0.909, *p* = 0.143, respectively) (Figure 10). 

The claudin-4 content of the cultured cells was significantly decreased in the case of all of the applied NE treatments compared with the control cells (*p* = 0.020, *p* = 0.013, *p* = 0.003, respectively) (Figure 11). 

Significant decline in TER values was measured in the case of 100 and 1000 µM NE treatment, both immediately after the treatment (*p* = 0.04, *p* = 0.03, respectively) and after 24 h regeneration time (*p* = 0.038, *p* = 0.037, respectively) compared with the controls, but there was no significant decrease after the 10 µM NE exposure (*p* = 0.07, *p* = 0.057 for immediate and 24-h measures respectively) (Figure 12).

## 4. Discussion

In the present study, a novel primary uroepithelial cell culture of feline origin was successfully established as an in vitro model suitable for investigating the pathomechanism of FIC. To the best of our knowledge, this model is the first cell culture from a cat which has been proven to contain differentiated uroepithelial cells, and the origin of the cells has been verified by the immunofluorescent detection of uroplakin III. As uroplakins are specific differentiation products, they are suitable to characterize differentiated uroepithelial cells in cultures [8]. Cytokeratins are characteristic of epithelial cells, therefore, the positivity confirms the epithelial origin of the cultured cells [6,7,9]. The cytokeratin positivity of the cells could be detected from the first day after plating, but the uroplakin III positivity could be observed only on the sixth day after seeding, which may suggest that several days are required to obtain differentiated uroepithelial cells in cell cultures.

Based on earlier studies it can be stated that FIC is a complex, multifactorial disease, developed on the basis of the interactions between the urinary bladder, adrenal glands, and nervous system, also affected by the environment where the cats live [1]. Therefore, to understand the pathomechanism of the disease, our cell culture model could be beneficial as an in vitro system, in which the above-mentioned interactions and factors can be studied separately as well as in targeted combinations. Further, it should be highlighted that it is a non-tumorigenic primary cell culture, hence the results can be better extrapolated to the in vivo conditions of the feline urinary bladder than those obtained on cell lines.

The other main goals of the present study were to investigate the cellular effects of NE in the established cell culture model, a hormone which could be a major contributor in the pathomechanism of FIC [17], and, further, to examine the direct link between the release of stress hormones and inflammation, oxidative stress, and the barrier function of the cat’s bladder. According to previous studies, in cats with FIC, the physiological stress response system is remarkably altered, hallmarked by the permanent increase in the plasma norepinephrine level in a rest state as well [13,14]. Notwithstanding that norepinephrine is an important stress mediator in all domestic animal species, evaluating the effects of NE in cats is of special importance due to the aforementioned endocrine alterations in FIC-affected cats. NE acts on the alfa-1 adrenergic receptors in the smooth muscle layer and uroepithelial cells of the bladder wall; hence, this hormone can have a significant effect on the stress responses of the organ [26]. The applied NE concentrations were set to cover a wide range and based on the limited available data from previous studies [27]. The treatment procedure (1 h NE exposure followed by 24 h regeneration) was set to mimic acute stress with the appropriate time for triggering pro-inflammatory humoral response. Under in vivo conditions, a similar acute stressor can exacerbate the clinical signs of FIC in cats or those of interstitial cystitis in women by increasing the permeability of the bladder wall [14,28].

The metabolic activity of the cultures was monitored by a CCK-8 test. This assay is based on a colorimetric method and is suitable for measuring the real-time aerobic catabolic activity of cell cultures. After 1 h of 1000 µM NE treatment, the metabolic activity was significantly higher than in the non-treated control group. In the case of 10 and 100 µM NE treatments, the metabolic activity of the cells was not elevated at all. An increased metabolic activity may suggest the increased catabolic adaptation of cells exposed to NE induced stress. These results are consistent with those of previous studies, where the metabolic activity of a human osteoblast cell line was elevated after NE treatment [29]. Further, the increased metabolic activity of chicken hepatocytes grown in primary cultures was measured by a CCK-8 test after heat stress, referring to the rapid metabolic stress adaptation of the cells [30]. Cell injury was examined by measuring the extracellular LDH activity, and according to our results, that was not affected by NE, which indicates that the applied treatments were not cytotoxic and did not induce the cell damage of the cultured cells.

In the present study, the inflammatory effect of NE was examined by measuring the IL-6 and SDF-1 concentrations of cell culture media and cell lysate. In the cell culture media, the IL-6 concentration was elevated after 1 h of 1000 µM NE treatment, but there were no significant changes in the case of 10 and 100 µM NE application. However, in cell lysate, no significant changes could be measured after any treatments, which reflects that the cells secrete this cytokine into their environment. In a previous study, the effect of NE was also examined on the production of various cytokines, such as IL-6 on human immortalized gastric epithelial cell culture, and it was found that NE triggered increased IL-6 production and up-regulated the IL-6 receptor, confirming the role of NE in inflammatory processes [27]. These results are also in line with a previous study, in which the stimulatory effect of NE was detected on IL-6 production in rat cardiac fibroblast cells [31]. The enhanced pro-inflammatory cytokine release in urinary bladder disorders was also reported by previous studies, in which a significant increase in IL-6 levels was measured in the urine of women with interstitial cystitis, in rats with experimentally induced cystitis, and in cats with FIC [19,32,33].

A significant increase in the SDF-1 concentration in cell culture media was measured at all NE concentrations used, but there were no changes in the case of cell lysate, similarly to the IL-6 measurements. As a pro-inflammatory chemokine, the roles of SDF-1 and its receptor have been studied in rats with experimentally induced cystitis, where an increased blood SDF-1 concentration, as well as an elevated uroepithelial SDF-1 receptor expression, were observed [20]. In addition to the above-mentioned experiment, other studies have addressed the role of this chemokine in the pathomechanism of FIC: an elevated SDF-1 concentration was measured in the blood of cats with FIC [21], and in the urine of women with interstitial cystitis [34]. Therefore, taking every detail into consideration, it can be suggested that this chemokine, similarly to IL-6, may also have a role in the development of the disease. Based on our results, it should be underlined, that the pro-inflammatory chemokine SDF-1 proved to be more sensitive to the presence of NE than IL-6. This finding corresponds to those of Parys et al. (2018), reporting a remarkable increase in the plasma SDF-1 concentrations of FIC-affected cats, while plasma IL-6 levels remained unchanged.

To investigate if NE is able to induce oxidative stress in uroepithelial cells, certain oxidative stress markers (MDA, 8-OHdG, PC, GRP78) were monitored. According to our results, it could be concluded that the NE had a significant effect only on the concentration of 8-OHdG and PC among the assessed stress markers. The marker 8-OHdG is a common and sensitive biomarker of oxidative DNA damage [35]. Interestingly, in case of 100 µM NE treatment, the concentration of 8-OHdG was significantly decreased, however some previous papers highlighted that hormonal stress is able to induce oxidative DNA damage (reflected by an elevated 8-OHdG level) in human oral keratinocyte cells, and also in hepatocytes of rats [36,37]; further, an increased 8-OHdG level was detected in the urine of patients with interstitial cystitis [38,39]. Based on these previous data, the above-mentioned paradoxical effect may be explained rather by the over-compensating processes of uroepithelial cells, than the protective role of NE against oxidative stress.

PC measurement is a useful method to investigate the protein damage caused by oxidative stress [40]. In this present study, the concentration of PC was elevated in the case of the highest concentration (1000 µM) of NE treatment, in accordance with a previous study, in which increased stress hormone release induced a slightly elevated PC level in cultured rat urothelial cells [41]. 

MDA is an oxidative stress marker resulting from lipid peroxidation [42]; further, GRP78 is a stress related endoplasmic reticulum (ER) chaperon protein, which could also serve as an oxidative stress marker [43]. Some earlier papers showed an increased urinary MDA level in human interstitial cystitis [39], elevated GRP78 expression in uroepithelial cells of rats with cystitis [44], and also an increased GRP78 level in different cell types after NE treatment [45,46]. However, in the present study, both the MDA and GRP78 levels remained unchanged. It seems that the applied short-term NE treatment was not able to cause significant DNA damage, lipid peroxidation or ER distress in cultured feline uroepithelial cells.

Another goal of the present paper was to investigate the effect of stress on the barrier function of the cultured uroepithelial cells; hence, the GAG concentration of the cell culture media, claudin-4 content of cultured cells, and TER of the layer of cultured epithelial cells were investigated after NE exposure. In healthy cats, the apical surface of bladder urothelium is covered by a GAG layer which may have an important role in maintaining the barrier function [11]. Certain previous studies highlighted some changes in the bladder GAG layer, such as the declined total urinary GAG concentration and decreased abundance of a specific GAG, GP-51, in the bladder of cats suffering from FIC, although it is still unclear whether this decrease is due to the failure of GAG synthesis or metabolism [11,47,48]. These findings are in line with our results, showing that the 100 µM NE treatment has significantly decreased the concentration of GAGs in cell culture media, indicating that NE may impair physiological GAG metabolism. 

To investigate the barrier function of the uroepithelium, it is inevitable to examine the tight junction proteins as well, since the barrier function is supported by the structural integrity of tight junctions [49]. Claudins form a multigene protein family with 24 members, being able to form ion channels and to regulate the membrane permeability depending on the size, charge, and electrical resistance of the ions [50]. Claudin-4, -8, and -12 can be found in tight junctions of the umbrella cells [49]. Previous studies have already indicated alterations in the abundance and distribution of certain tight junction proteins, such as zonula-occludens 1, occludin, E-cadherin, and uroplakins in cats with FIC and human patients with interstitial cystitis [2,51,52]. In this present study, all the applied NE treatments (10-, 100-, and 1000 µM) had a significant decreasing effect on the claudin-4 concentration of the cultured cells, indicating that the NE-triggered stress can diminish the barrier function of urothelial cells. Similarly, 100 and 1000 µM NE exposures had a decreasing effect on the TER of the uroepithelial monolayer, both immediately after treatment and following 24 h regeneration time compared to the non-treated control cells. These findings are in line with an earlier study, where reduced TER was measured in the bladder of cats suffering from FIC [2]. Taken together, these results demonstrate that NE has a direct influence on uroepithelial barrier functions, which may be in association with changes in the expression of tight junction protein claudin-4 and the GAG content of urothelial cells.

To the best of our knowledge, stress-associated idiopathic cystitis is relevant mostly in cats and humans [1,2]. In contrast, in other companion animals such as dogs, the similar symptoms are most often caused by a bacterial urinary tract infection, and only the minority of cases remain idiopathic [53,54,55]. Hence, monitoring the effects of stress on uroepithelial cells in further species, such as dogs, is presumably not as relevant as in cats. However, it cannot be excluded that norepinephrine exposure could similarly affect bladder epithelial cells of other species, which should be addressed in future studies.

According to our recent knowledge, this was the first study which investigated the direct molecular effect of NE on cultured feline uroepithelial cells to gain a deeper knowledge of the pathomechanism of FIC; however, for a more accurate understanding of the disease, further research is necessary, and the established novel cell culture can serve as a proper model for studying the molecular background of the illness in the future.

## 5. Conclusions

NE was able to increase the metabolic activity of the cultured uroepithelial cells, suggesting that the enhanced cellular metabolism might have an important role in the stress response of the organ. Further, the short-term NE treatment could increase the SDF-1 production of uroepithelial cells, even at lower concentration; however, NE-triggered IL-6 secretion was observed only after applying higher concentrations. NE was able to induce the oxidative protein damage of cultured urothelial cells, but no DNA damage, lipid peroxidation, or ER stress was detected in the case of the applied short-term NE treatments. As a result of the applied acute stress, the barrier function of feline uroepithelial cells was damaged, as reflected by the reduced TER values, which may be in association with the reduced expression of claudin-4 tight junction protein and the decreased GAG concentration.

These results suggest that NE as a stress hormone may have pro-inflammatory action, can contribute to oxidative protein damage, and may impair the barrier function of the urothelium, hence it may play an important role in the pathogenesis of the FIC as well. Further, the established feline primary uroepithelial cell culture can be considered as a proper in vitro model for studying the molecular mechanisms of urinary disorders in cats.

## Figures and Tables

**Figure 1 vetsci-10-00132-f001:**
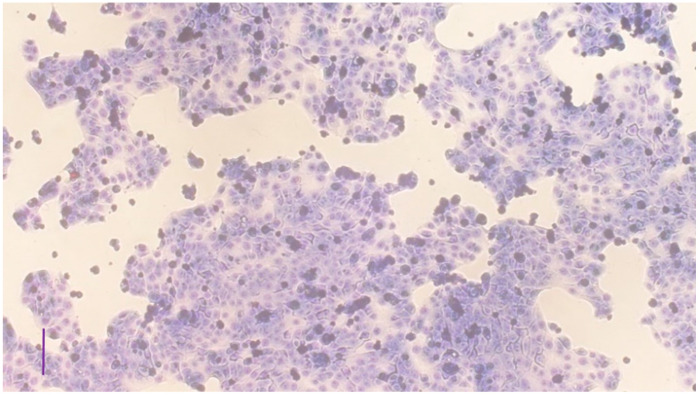
Uroepithelial cell culture on day 1 after plating stained by Giemsa (bar = 100 µm).

**Figure 2 vetsci-10-00132-f002:**
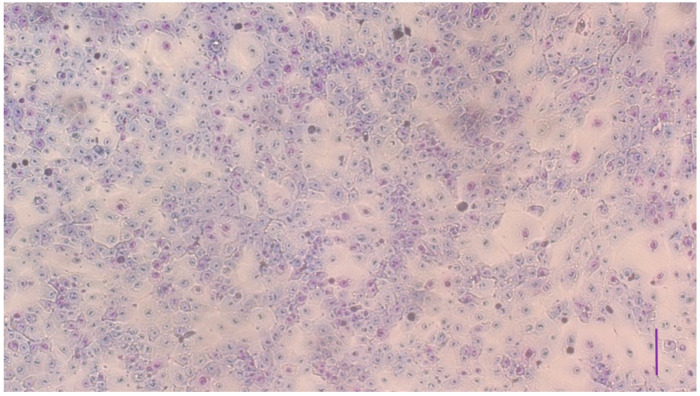
Confluent uroepithelial cell culture on day 6 after plating stained by Giemsa (bar = 100 µm).

**Figure 3 vetsci-10-00132-f003:**
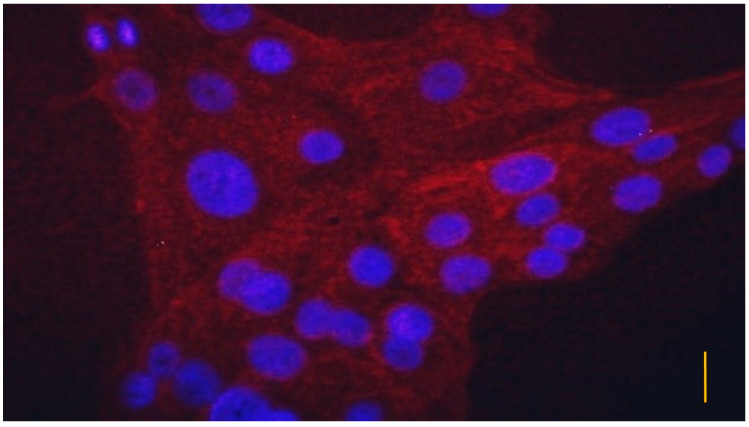
Immunofluorescent staining of uroepithelial cell culture with eFluor labeled pan-cytokeratin antibody on day 1 after plating. Blue color indicates diamidino phenylindole (DAPI) labeled cell nuclei, while red color shows epithelial cells detected with eFluor labeled pan-cytokeratin antibody. (bar = 3.5 µm).

**Figure 4 vetsci-10-00132-f004:**
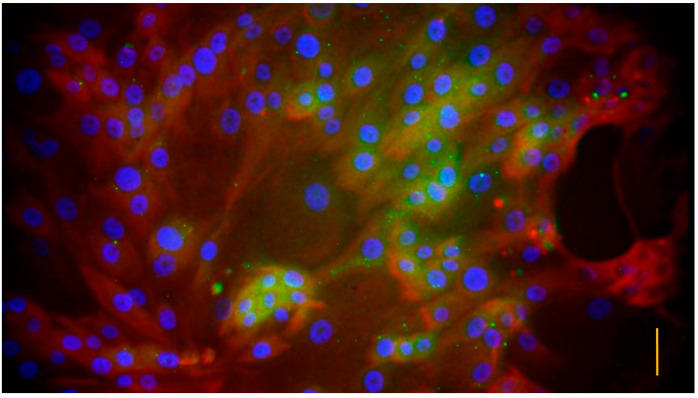
Immunofluorescent staining of uroepithelial cell culture on day 6 after plating. Blue color indicates diamidino phenylindole (DAPI) labeled cell nuclei, red color shows epithelial cells detected with eFluor labeled pan-cytokeratin antibody, and green color refers to differentiated uroepithelial cells stained with fluorescein isothiocyanate (FITC) labeled uroplakin III antibody. (bar = 3.5 µm).

**Figure 5 vetsci-10-00132-f005:**
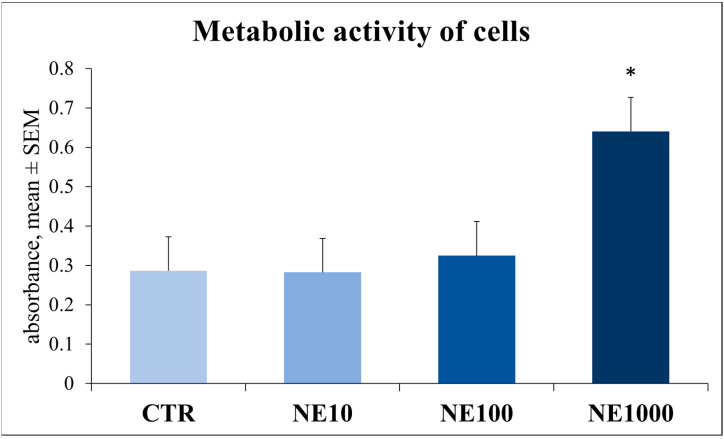
Metabolic activity of cells after 1-h norepinephrine treatment. CTR = non-treated control cells, NE10 = 10 µM norepinephrine treated cells, NE100 = 100 µM norepinephrine treated cells, NE1000 = 1000 µM norepinephrine treated cells. *n* = 5/group, results are expressed as mean ± SEM, * *p* < 0.05.

**Figure 6 vetsci-10-00132-f006:**
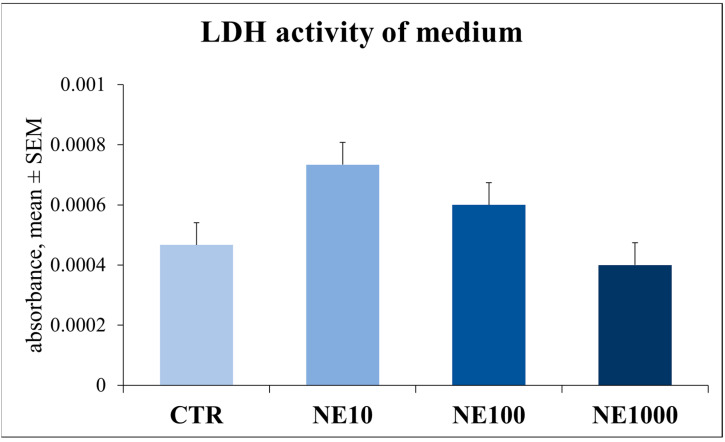
Lactate dehydrogenase (LDH) activity in cell culture media after 1-h norepinephrine treatment. CTR = non-treated control cells, NE10 = 10 µM norepinephrine treated cells, NE100 = 100 µM norepinephrine treated cells, NE1000 = 1000 µmol/L norepinephrine treated cells. *n* = 6/group, results are expressed as mean ± SEM.

**Figure 7 vetsci-10-00132-f007:**
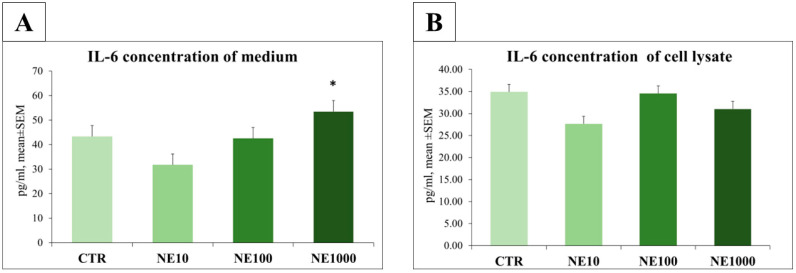
Interleukin-6 (IL-6) concentrations in culture media (**A**) and cell lysate (**B**) after 1-h norepinephrine (NE) treatment. CTR = non treated control cells, NE10 = 10 µM norepinephrine treated cells, NE100 = 100 µM norepinephrine treated cells, NE1000 = 1000 µM norepinephrine treated cells. *n* = 6/group, results are expressed as mean ± SEM, * *p* < 0.05.

**Figure 8 vetsci-10-00132-f008:**
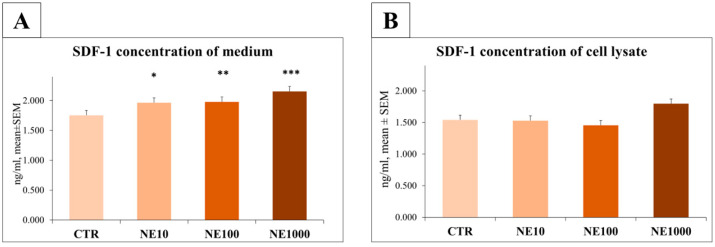
Stromal-cell derived factor-1 (SDF-1) levels in culture media (**A**) and cell lysate (**B**) after 1-h norepinephrine (NE) treatment. CTR = non-treated control cells, NE10 = 10 µM norepinephrine treated cells, NE100 = 100 µM norepinephrine treated cells, NE1000 = 1000 µM norepinephrine treated cells. *n* = 6/group, results are expressed as mean ± SEM, * *p* < 0.05, ** *p* < 0.01, *** *p* < 0.001.

**Figure 9 vetsci-10-00132-f009:**
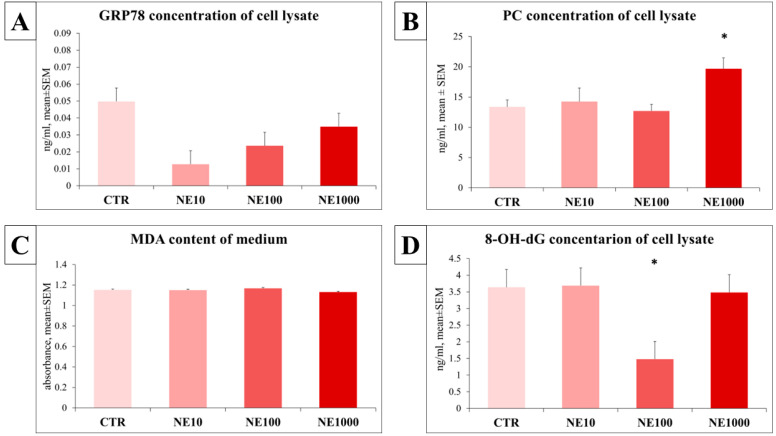
Concentration of oxidative stress markers after 1-h norepinephrine (NE) treatment. Glucose-regulated protein 78 (GRP78) (**A**) and protein carbonyl (PC) (**B**) concentrations in cell lysate, malondialdehyde (MDA) concentration of cell culture media (**C**), and 8-hydroxy 2 deoxyguanosine (8-OHdG) level of cell lysate (**D**). CTR = non-treated control cells, NE10 = 10 µM norepinephrine treated cells, NE100 = 100 µM norepinephrine treated cells, NE1000 = 1000 µM norepinephrine treated cells. *n* = 6/group, results are expressed as mean ± SEM, * *p* < 0.05.

**Figure 10 vetsci-10-00132-f010:**
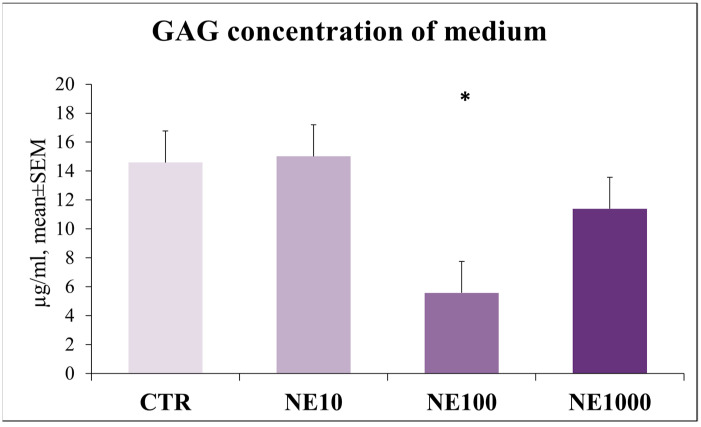
Glycosaminoglycan (GAG) concentration of culture medium after 1-h norepinephrine (NE) treatment. CTR = non-treated control cells, NE10 = 10 µM norepinephrine treated cells, NE100 = 100 µM norepinephrine treated cells, NE1000 = 1000 µM norepinephrine treated cells. *n* = 6/group, results are expressed as mean ± SEM, * *p* < 0.05.

**Figure 11 vetsci-10-00132-f011:**
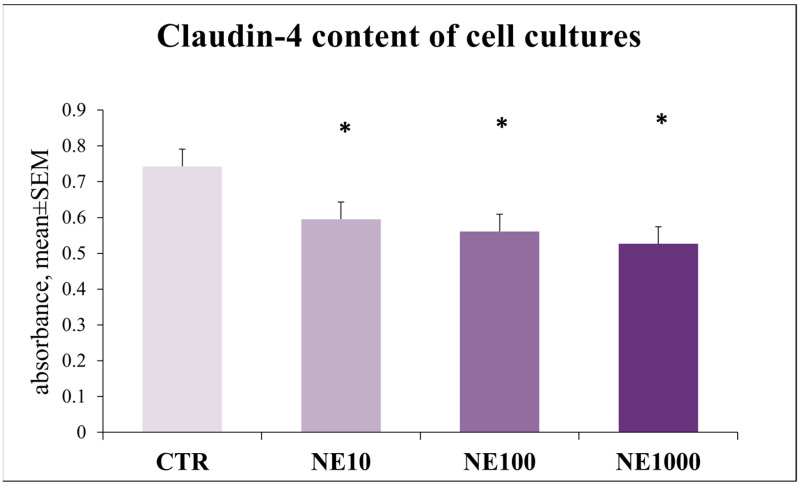
Claudin-4 content of uroepithelial cell culture after 1-h norepinephrine (NE) treatment. CTR = non-treated control cells, NE10 = 10 µM norepinephrine treated cells, NE100 = 100 µM norepinephrine treated cells, NE1000 = 1000 µM norepinephrine treated cells. *n* = 6/group, results are expressed as mean ± SEM, * *p* < 0.05.

**Figure 12 vetsci-10-00132-f012:**
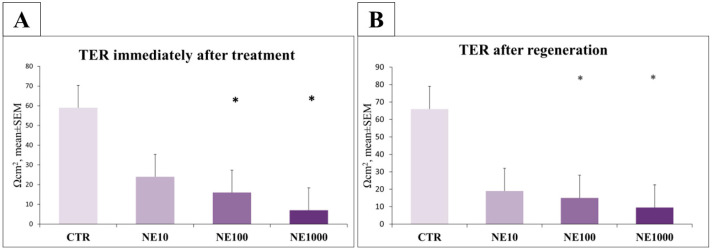
Transepithelial electrical resistance (TER) of uroepithelial cell cultures measured immediately after norepinephrine treatment (**A**) and after 24 h regeneration time (**B**). CTR = non-treated control cells, NE10 = 10 µM norepinephrine treated cells, NE100 = 100 µM norepinephrine treated cells, NE1000 = 1000 µM norepinephrine treated cells. *n* = 3/group, results are expressed as mean ± SEM, * *p* < 0.05.

## Data Availability

All raw data supporting the results of the present study can be obtained from the corresponding author upon reasonable request.

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
