# Peer review of "Feline Uroepithelial Cell Culture as a Novel Model of Idiopathic Cystitis: Investigations on the Effects of Norepinephrine on Inflammatory Response, Oxidative Stress, and Barrier Function"

_vetsci, 2023, doi:10.3390/vetsci10020132_

Round 1

Reviewer 1 Report

Feline idiopathic cystitis (FIC) is one of the critical urinary disorders in cats, and its pathophysiological mechanisms have not fully been clarified. It is regarded that this report contains important data to clarify the unknown mechanisms of the FIC, and this manuscript is basically well-written. However, several points should be revised for publication.

 Major comments

1. The analysis of the present study was done with ELISA and immunofluorescence. In these techniques, confirmation of the species-specificity of the antibodies and kits is very important to guarantee the accuracy of the data. However, there are no descriptions regarding this issue. The authors should show evidence that the antibodies and ELISA kits used in the present study can certainly detect the substances of feline samples.

2. L525. Institutional Review Board Statement: Although the samples were obtained from a client-owned cat with permission of the owner, the experiments should be done after permission of the appropriate committee of the institute.

 3. The effects of norepinephrine in the cultured uroepithelial cells demonstrated in the present study might be one of the mechanisms of the FIC. However, it is unknown if these events are feline-specific or common to other species like dogs. Is there a possibility that similar responses by norepinephrine of epinephrine to the uroepithelial cells also occur in other species? Please make discussion this issue.

Minor comments

1. L45: Lower urinary tract dysfunction→Lower urinary tract disease

Reviewer 2 Report

Dear all,

The present manuscript described The paper was well written, concise, and well described.

The present results showed that treatment with norepinephrine exposure affected uroepithelial cells by increasing metabolic activity, triggering oxidative stress and decreasing the barrier integrity of the cultured cells. These results suggest that norepinephrine as a stress hormone may have pro-inflammatory action, can contribute to oxidative protein damage, and may impair barrier function of the urothelium, hence it may play an important role in the pathogenesis of the FIC as well.

Therefore, these results may be useful for clinicians as well as new therapies against urinary disorders of cats.

Minor suggestions are below:

- line 407: I suggest to change "necrosis" to "cell damage", since in my undestanding the authors di not provide data from experiments differentiating apoptosis from necrosis 

- line 499: I suggest you remove "Summarizing the result of the present study, it can be concluded that"

Round 2

Reviewer 1 Report

There are no comments. I confirmed that the author adequately revised the manuscript.